# OpenReview forum: "Memory Self-Regeneration: Uncovering Hidden Knowledge in Unlearned Models"
_ICLR.cc/2026/Conference — Submitted to ICLR 2026_

### Official Review · Reviewer_NMoc · 2025-10-26

**Soundness:** 2
**Presentation:** 2
**Contribution:** 2
**Rating:** 6
**Confidence:** 3

**Summary:**

This paper introduces Memory Self-Regeneration (MSR), a novel framework for analyzing how unlearned diffusion models can “self-recover” forgotten concepts. The proposed method, MemoRa (Memory Regeneration with LoRA), combines DDIM inversion, spherical interpolation, and LoRA fine-tuning to regenerate erased concepts using only a few images. The authors also introduce the notion of short-term and long-term forgetting to model differences in recoverability. Experiments demonstrate that several existing unlearning methods fail to truly erase knowledge.

**Strengths:**

1. Prior work mainly focuses on whether forgetting is achieved, not whether forgotten concepts can be reactivated. This paper drawing analogies between human short/long-term memory and model forgetting offers a conceptually rich and interdisciplinary framing.

2. Quantitative results (FID, CLIP, NudeNet detection, ASR) and visualizations consistently show that MemoRa can rapidly restore “forgotten” concepts, verifying the persistence of hidden knowledge.

3. The study addresses a crucial issue in model safety and compliance — whether “machine unlearning” truly eliminates harmful or private content.

4. The paper is well-organized and clearly written. Figures (especially Fig. 1–2) effectively illustrate conceptual and procedural flow.

**Weaknesses:**

1. The paper primarily demonstrates empirical findings but lacks a formal analysis explaining why the model retains recoverable memory at a representational level (e.g., through weight-space or manifold topology).

2. It remains unclear whether MemoRa reveals true residual memory or simply relearns from scratch given minimal samples. The “self-regeneration” claim would benefit from ablation comparing to training from random initialization.

3. While the paper introduces short- vs long-term forgetting qualitatively, it does not propose a quantitative metric to measure the degree of residual knowledge.

4. Experiments focus on Stable Diffusion v1.4; extension to other architectures (e.g., text-to-video, large-scale T2I foundation models) would strengthen generality.

**Questions:**

See Weaknesses

---

> ### Author Response · Authors · 2025-11-22
>
> We appreciate the reviewer’s valuable feedback and are encouraged by their recognition of our work’s potential, particularly in highlighting the importance of investigating whether “machine unlearning” can genuinely remove harmful or sensitive information in the context of model safety and compliance.
>
> **W1:** A  deeper **conceptual and representation-level analysis was added** in Section 4. In this section, we examine how different unlearning strategies affect the latent manifold and model features using FID scores and PCA-reduced visualizations. Our analysis offers insight into the mechanisms underlying short-term and long-term forgetting and helps explain why residual knowledge remains recoverable in practice.
>
> **W2:** **To address this concern, we have added a new experiment (see Tab. 4 )** in which we perform targeted recovery on a model that was unlearned for nudity using only one subset of examples (e.g., female images) and then evaluate recovery on the complementary subset (e.g., male images), and vice versa.  The results show that the model can recover nudity concepts for the unseen subset, demonstrating that MemoRa restores residual knowledge rather than simply relearning from scratch. This provides empirical support for the “self-regeneration” claim and indicates that the model retains underlying concept representations that can be reactivated.
>
> **W3:** We agree with the Reviewer that the original manuscript did not provide a quantitative metric to assess residual knowledge. In response, **we introduce a Knowledge Recovery Score (KRS) that quantifies how effectively a method restores forgotten knowledge**. We introduce a new measure formulated as a Knowledge Recovery Score (KRS), calculated as follows:
>
> KRS = (Perf(MemoRa) - Perf(Unlearned)) / (Perf(Base) - Perf(Unlearned))
>
> Where:
> - Perf() is the accuracy of the classifier (e.g. ResNet, NudeNet Detector) that checks whether the generated image contains a specific concept.
> - Perf(Base) is the performance of the original model on the target concept.
> - Perf(Unlearned) is the model's performance after the unlearning process.
> - Perf(MemoRa) is the performance after applying the MemoRa recovery process.
>
> A score close to 1.0 would indicate that the concept was fully recovered (short-term forgetting), while a score close to 0.0 would suggest the concept was permanently erased (long-term forgetting). This metric would allow for a systematic and measurable analysis of the effectiveness of various unlearning techniques, see Tab. 3.
> Thank you for the suggestions. We believe that adding this metric will significantly improve the clarity and depth with which we address a topic of substantial importance.
>
> **W4**: In the manuscript, we followed commonly used settings from the current Machine Unlearning literature (Gandikota et al., 2023). We agree with the Reviewer, however, that diffusion models have advanced substantially, and using older architectures like SD-1.4 may not fully reflect the behavior of modern models such as SDXL and FLUX. In this rebuttal, **we have added new results specifically for the FLUX architecture, as shown in the updated teaser (Fig. 1) and in Figures 3, 4, and Tables 6 and 13**.
> It is worth noting that for Flux-based models, the use of memory not only restores the underlying concept but also produces images that closely resemble the reference examples (as is particularly evident for UCE in Figure 4, where the desert, cliffs and river return very well). We trust that these results demonstrate that our findings remain relevant for modern architectures.

---

> > ### Author Response · Authors · 2025-11-27
> >
> > Thank you again for your constructive feedback. In the rebuttal we considered your feedback and responded to all the points raised, including the new Knowledge Recovery Score (KRS) and evaluation on modern architectures like FLUX. Please let us know if you need any additional clarifications, we would be happy to address them.

---

### Official Review · Reviewer_4P9M · 2025-10-27

**Soundness:** 2
**Presentation:** 3
**Contribution:** 2
**Rating:** 2
**Confidence:** 4

**Summary:**

This paper investigates the phenomenon that diffusion models may retain residual knowledge even after being “unlearned.” The authors propose a new evaluation task, Memory Self-Regeneration (MSR), and a corresponding LoRA-based method called MemoRa. The idea is to show that a few samples can “re-trigger” previously removed concepts from an unlearned model. The paper further introduces a distinction between “short-term forgetting” and “long-term forgetting,” drawing an analogy with human memory mechanisms.

**Strengths:**

The paper explores an interesting topic, forgetting recovery, which is important for safety and compliance of diffusion models.

**Weaknesses:**

1. Regarding the definition of short-term and long-term forgetting. I don't think this can be defined as short-term and long-term forgetting. When unlearning a diffusion model, if the number of forgetting training steps is sufficient, the model parameters deviate significantly from their initial positions, resulting in forgetting that is difficult to recover. Conversely, if the number of forgetting training steps is small or only the inference process is modified, the movement of model parameters is minimal, making the forgetting easier to recover.
2. Regarding MemoRa, fine-tuning is performed using datasets created with Lora and interpolation. Interpolation increases the number of images related to the concept, and as the volume of image data for the relevant concept grows during the training process, the model becomes increasingly likely to revert to its state before forgetting the concept. This is merely one way of generating data, with a relatively minor contribution.
3. The level of FID indicates the degree to which the model has forgotten its training. Therefore, can FID reflect the extent of "memory recovery"?
4. It appears that the use of spherical interpolation is no different from ordinary latent data augmentation, but the authors did not explain why they chose this method, nor did they provide ablation comparisons.

**Questions:**

See weaknesses.

**Details Of Ethics Concerns:**

No.

---

> ### Author Response · Authors · 2025-11-22
>
> We thank the Reviewer for the feedback and constructive remarks regarding our paper that we believe will improve our paper.
>
> It is very important to us that the reviewer notes that the paper explores an interesting topic, forgetting recovery, which is important for the safety and compliance of diffusion models. We would like to add that we consider this to be very important because we show the imperfection of unlearning, indicating the need to develop this topic further
>
> **W1**: We maintain that the distinction between short-term and long-term forgetting is meaningful in diffusion models, as these regimes reflect different dynamics in how knowledge is erased and can be recovered. We agree with the Reviewer, however, that the initial manuscript did not provide sufficient empirical justification for this claim. **In response to this need, we have added a dedicated section titled "Short- vs. Long-Term Forgetting" (Section 4), where we analyze these regimes using quantitative representation-level measurements.**
> Specifically, we evaluate each unlearning method by computing the FID score between the distributions of features produced by Stable Diffusion and those produced by the unlearned model. Methods we classify as short-term exhibit only minimal displacement according to FID, whereas long-term methods produce substantially larger shifts. This allows us to operationalize these two regimes based on measurable divergence rather than qualitative interpretation.
> To further illustrate the geometric nature of these differences, we provide visualizations of the latent feature distributions in a PCA-reduced space. We also emphasise that the definition of short-term refers to the pair of model and concept, not just to the model. We hope this additional explanation helps you better understand the paper.
>
>
> **W2**: We would like to stress that the simplicity of the method is a deliberate design choice. **Our core contribution lies in showing that even a minimal and efficient strategy**, absent advanced algorithmic components, can successfully mitigate the impact of current unlearning techniques
>
> The paper's focus is on revealing the vulnerability of existing unlearning methods, showing that many of them only achieve a superficial, short-term forgetting that is easily reversible. As we believe that the topic of data protection and unlearning is important, in our paper we raise the issue of the easy restoration of data that has not been completely forgotten.
>
>
> **W3**: We use FID score (between the original dataset and the results of learned models) to measure how much the model has forgotten to generate the general concepts, not the unlearned concept. A significant increase in the FID score after unlearning indicates that the method has harmed the model's overall image generation quality, suggesting that the unlearning caused the model to leave the original representation manifold. Consequently, an increase of FID score after the MemoRa application indicated the recovery of the model's general generation quality. **The actual recovery of the unlearned concept is measured by other metrics like Attack Success Rate (ASR) and concept-specific detectors (like NudeNet). Therefore, FID score can’t be used to assess the concept recovery, but only to check the similarity of general concept generation.** We hope this explanation clarifies the specific role of FID in our evaluation framework.
>
> **W4**: This choice is a standard approach for navigating the latent spaces of generative models [1, 2]. The underlying idea is that the latent space is often structured as a hypersphere, where the most plausible data points lie on its surface. By using slerp, the training set can be efficiently generated from a small number of images. **A visualisation showing the comparison of the two interpolations is presented in Fig. 7, and the numerical results in Tab. 7**. Thank you for this observation. We trust that the inclusion of supplementary references and visualisations has made our methodological choices clearer to the reader.
>
> [1] Davidson, T. R., Falorsi, L., De Cao, N., Kipf, T., and Tomczak, J. M. (2018). Hyperspherical Variational Auto-Encoders. 34th Conference on Uncertainty in Artificial Intelligence (UAI-18)
>
> [2] White, T. (2016). Sampling Generative Networks.

---

> > ### Author Response · Authors · 2025-11-27
> >
> > Thank you again for your constructive feedback. We carefully addressed all remarks in the rebuttal and the updated manuscript, including additional ablation studies, and clarifying the difference between short- and long-term forgetting. Do you have any further comments or concerns we could still address? We would be glad to incorporate any remaining improvements.

---

> > > ### Comment · Reviewer_4P9M · 2025-11-28
> > >
> > > Thanks for your rebuttal. I appreciate the authors’ efforts in addressing my concerns. However, it appears that the system currently does not allow me to modify my score. If it becomes possible later, I will update my score.

---

### Official Review · Reviewer_Axny · 2025-10-30

**Soundness:** 3
**Presentation:** 3
**Contribution:** 3
**Rating:** 8
**Confidence:** 3

**Summary:**

The manuscript studies memory self-regeneration, focused on  analyzing knowledge recovery mechanisms in models, with particular
emphasis on their ability to recall information that has been previously unlearned. The proposed method aims to recover unlearned
information using only a few images containing removed concepts.

Two distinct modes of model forgetting are studied: a short-term form, where concepts can be quickly recalled, and a long-term  form, where recovery is slower and demanding.

The manuscript aims to answer the question whether the unlearned diffusion models are capable of self-regenerating forgotten information?

The manuscript proposes MemoRa, a strategy for recalling knowledge in unlearned models, with a particular focus on approaches based
on Low-Rank Adaptation (LoRA). Spherical interpolation is used to expand the latent space in order to generate more relevant images.

**Strengths:**

* The manuscript provides a study in models being able to recall information after unlearning.
* It provides a simple solution for recalling past information.
* It provides extensive experimental results.

**Weaknesses:**

* The manuscript lacks a theoretical justification that guarantees the recovery of information or provides bonds whether the information can
be recovered.

**Questions:**

What happens that unlearning and relearning are repeated, like in a process of continual learning?

---

> ### Author Response · Authors · 2025-11-22
>
> We would like to thank the Reviewer for understanding and supporting our work in an important topic which is recall information after unlearning.
>
> **W1**: The concept of memory regeneration of diffusion models is complex, and consequently creating a theoretical framework is challenging. We build our findings on the concepts of exact and approximate unlearning discussed in the related literature. Our proposed method can be viewed as a technique for validating and reversing the current approximate methods. Instead, in the revised manuscript, **we provide a deeper empirical and conceptual analysis of the forgetting dynamics**, presented in Section 4, which examines how different unlearning strategies affect the latent manifold and the accessibility of erased knowledge. This analysis offers insight into the mechanisms underlying short-term and long-term forgetting, even in the absence of formal theoretical bounds. We believe this empirical analysis provides a solid foundation for understanding the underlying dynamics.
>
> **Q1**: We thank the Reviewer for this interesting question. While our current study focuses on single-shot unlearning and recovery, the dynamics of repeated unlearning and relearning, akin to continual learning are an important direction for future work. Based on our analysis in Section 4, we hypothesize that repeated cycles could amplify differences between short-term and long-term forgetting: short-term forgetting may remain largely reversible across iterations, whereas long-term forgetting could accumulate structural changes in the latent manifold, making recovery progressively more difficult. Investigating such effects systematically would require tracking latent manifold shifts and Knowledge Recovery Scores across multiple unlearning cycles, which we leave as future work.  It would also be worthwhile to examine scenarios in which greater degrees of forgetting correspond to improved performance of the recovery algorithm. We appreciate this suggestion as it identifies an interesting research direction.

---

> > ### Author Response · Authors · 2025-11-27
> >
> > Thank you for your positive initial review and acknowledgement of the strengths of our work.
> > We are grateful for the reviewer’s understanding and support of our contribution to this important topic. If you have any other questions, we would be happy to address them.

---

### Official Review · Reviewer_7yCy · 2025-11-01

**Soundness:** 2
**Presentation:** 3
**Contribution:** 2
**Rating:** 2
**Confidence:** 3

**Summary:**

This paper introduces Memory Self-Regeneration, a task for evaluating the robustness of machine unlearning methods in text-to-image diffusion models. The authors also propose MemoRa, a recovery strategy that uses DDIM inversion combined with LoRA to relearn supposedly erased concepts using only a few reference images. Through experiments on Stable Diffusion 1.4 with various unlearning methods, they demonstrate that many unlearning approaches exhibit "short-term forgetting" where concepts are easily recovered, while others achieve "long-term forgetting" that is more resistant to recovery.

**Strengths:**

- The paper provides extensive experiments across multiple state-of-the-art unlearning methods (ESD, FMN, MACE, SalUn, AdvUnlearn, etc.) and diverse concept types (nudity, objects, artistic styles, celebrities), offering valuable insights into the brittleness of current approaches.
- The MSR task provides a concrete framework for evaluating unlearning robustness that could benefit the research community.
- The MemoRa strategy demonstrates that recovery can occur with minimal computational resources (only 6 images, ~15 minutes of LoRA training), which is an important finding for understanding real-world vulnerabilities of unlearned models.
- The AutoMemoRa variant shows consideration of practical trade-offs by addressing the FID degradation issue, and Multi-MemoRa demonstrates the approach's extensibility to multiple concepts.
- The evaluation setup is well-documented with appropriate metrics (FID, CLIP, NudeNet, ResNet-50 classification) and includes both standard generation and adversarial attack scenarios.

**Weaknesses:**

- Limited The proposed MemoRa strategy is a straightforward combination of existing techniques (DDIM inversion + spherical interpolation + LoRA fine-tuning) without algorithmic innovation. The paper does not explain why this specific combination is optimal or provide ablations justifying each component. Other works have explored more sophisticated approaches to memory extraction in diffusion models, such as textual embedding-based methods [1].
- All experiments are conducted exclusively on Stable Diffusion 1.4. The paper provides no evidence that findings generalize to modern, transformer-based architectures like SDXL, SD 3.0, FLUX or Sana.
This is a critical limitation given that architectural choices significantly impact both unlearning effectiveness and memory retention.
- The short-term vs. long-term memory analogy borrowed from neuroscience reduces to "easy to recover" vs. "hard to recover" in practice. The paper does not provide mechanistic insights into what causes these differences beyond speculation about manifold displacement (Section 3, lines 116-123). The hypothesis that short-term forgetting corresponds to "moving away from the manifold" while long-term forgetting reflects "displacement along the manifold" lacks empirical validation through representation analysis or ablation studies.
- The paper does not discuss recent work on quantization-based vulnerabilities in unlearned LLMs [2], which shows that forgotten knowledge can re-emerge even without explicit retraining, as well as alternative memory extraction approaches in diffusion models using textual embeddings [1] and maybe theoretical work on why approximate unlearning methods fundamentally struggle with complete erasure.
- Only LoRA-based recover is explored. Comparison with alternative approaches like full fine-tuning, embedding optimization or prompt tuning would strengthen claims about unlearning brittleness.
- The paper identifies a problem but offers limited guidance on how unlearning method developers should respond. Should they optimize directly for MSR robustness? If so, how? The conclusion states that MemoRa "struggles when the erased concept has been more deeply replaced" but doesn't explain how to achieve this "deeper replacement."


While this paper addresses an important question about the reliability of machine unlearning, it suffers from critical limitations and lacks technical novelty, since it is essentially an empirical observation rather than a methodological advance. Additionally, experiments are confined entirely to SD 1.4, providing no evidence of generalization to modern models or different architectural families. The short-term vs. long-term memory framing, while conceptually appealing, reduces to "easy vs. hard to recover" without mechanistic insights or predictive value.
In its current form, this represents a useful but incremental empirical study that highlights known limitations of approximate unlearning rather than providing significant new insights or techniques.


[1] Kowalczuk et al., "Finding Dori: Memorization in Text-to-Image Diffusion Models Is Not Local"
[2] Zhang et al., "Catastrophic Failure of LLM Unlearning via Quantization", ICLR 2025

**Questions:**

- Can you provide ablations on key design choices like number of seed images or DDIM inversion starting timestep (t=35 vs. other values)?
- The authors hypothesize that short-term forgetting involves "moving away from the manifold" while long-term forgetting involves "displacement along the manifold" (Section 3). Can you provide empirical evidence for this claim? Maybe using latent space visualization (t-SNE/UMAP of representations), distance to nearest training examples or analyzing intermediate layer activations.
- If MSR reveals that a method exhibits short-term forgetting, what should developers do? Can you propose modifications to unlearning algorithms that would make them more robust to MSR attacks?

---

> ### Author Response · Authors · 2025-11-21
>
> We sincerely thank the reviewer for their thoughtful comments and interesting questions. We are especially pleased that the overall the paper was well understood and the reviewer noted that our work is an important finding for understanding real-world vulnerabilities of unlearned models. We hope the responses below will address any remaining concerns and help us further improve the quality of the paper.
>
> **W1**: Thank you for your comment. We would like to emphasise that we wanted to show that the simplicity of the solution is an advantage, and **our main contribution is to show that even such a simple and computationally efficient approach**, without the need for advanced methods, can reverse the effect of current unlearning models. Furthermore, we show that more sophisticated approaches take longer and do not always perform better. Nevertheless, we show in our work that combining both methods (MemoRa and UnlearnDiffAtk) can often restore up to 100% of knowledge (see Tab. 12). It is worth noting that this is very dangerous in the context of system protection. The aim of our work was to point out a huge gap that had not been explored before. We hope this explanation clarifies the motivation behind our work.
>
> **W2**: In the manuscript, we used widely adopted settings from the current Machine Unlearning literature (Gandikota et al., 2023). However, we agree with the reviewer that diffusion models have evolved significantly, and the use of older architectures such as SD-1.4 may not be representative of the performance of modern architectures. **In the rebuttal, we included new results explicitly focused on the FLUX architecture, see updated teaser (Fig. 1) and Fig. 3, 4 and Tab. 6, 13**. We believe the additional results demonstrate that MemoRa is applicable to modern architectures.
>
> **W3**: We agree with the Reviewer that the initial version of the paper did not provide direct empirical evidence supporting the distinction between short-term and long-term forgetting. In response, we have added a dedicated section titled "Short- vs. Long-Term Forgetting" (Section 4), where we analyze these regimes using quantitative representation-level measurements.
> Specifically, we evaluate each unlearning method by computing the FID score between the distributions of features produced by Stable Diffusion and those produced by the unlearned model. **Methods we classify as short-term exhibit only minimal displacement according to FID, whereas long-term methods produce substantially larger shifts. This allows us to operationalize these two regimes based on measurable divergence rather than qualitative interpretation.**
> To further illustrate the geometric nature of these differences, we provide a visualization of the latent feature distributions in a PCA-reduced space (see Fig. 10). We trust the additional insights into the behaviour of forgetting provides a solid empirical basis for the proposed distinction.
>
> **W4**: We agree that recent findings on quantization-based vulnerabilities in unlearned LLMs [2] are relevant to the broader discussion of incomplete erasure. At the same time, we note that the mechanisms underlying forgetting and recovery in autoregressive language models differ substantially from those in text-to-image diffusion models. In LLMs, quantization can directly expose residual token-level statistical structure, whereas in diffusion models the dominant factors are latent-space geometry, denoising dynamics and visual feature priors. For this reason, results obtained in LLMs cannot be directly transferred to the diffusion setting.
>
> **W5**: We thank the Reviewer for the comment. Unlearning methods operate through different mechanisms, including weight modification, prompt tuning, or changes to the text encoder. LoRA-based recovery is particularly appealing because it provides a unified and lightweight interface that can be applied on top of all these strategies, including methods such as FLUX. By using LoRA, we can consistently study memory recovery across diverse unlearning approaches without modifying their underlying mechanisms, making it an effective and practical tool for our analysis of unlearning brittleness.
>
> To provide context and comparison, we include methods operating in a similar paradigm, such as prompt-tuning-based approaches exemplified by UnlearnDiffAtk. These methods rely on modifying prompt embeddings rather than model weights and therefore serve as a natural baseline for evaluating unlearning brittleness. We are confident that this our choice of LoRA as a tool for evaluating unlearning robustness.

---

> > ### Author Response · Authors · 2025-11-21
> >
> > **Q1**: The original submission used a fixed configuration of 6 reference images, expanded to 33 latent samples via spherical interpolation, and DDIM inversion with 50 steps. The updated version includes a full Tab. 5 of ablations regarding the number of reference images and the number of starting time step.
> >
> > **Q2**: We thank the Reviewer for this suggestion. In response, we have added an empirical analysis in Section 4 that examines short-term versus long-term forgetting. This includes latent space visualizations (PCA-reduced embeddings) and FID-based measurements comparing unlearned models to the original Stable Diffusion model, which we use to investigate and explain the nature of forgetting in these models, see Fig. 10. We hope this analysis provides valuable insights into the forgetting behaviours.
> >
> > **Q3/W6**:  We introduce the Knowledge Recovery Score (KRS), which quantifies how effectively a method restores forgotten knowledge. In updated Section 4, we describe the intuition behind short-term and long-term forgetting, relating it to the extent to which a method modifies the data manifold. Our analysis suggests that, for more robust unlearning, models should aim to modify the underlying manifold rather than merely breaking the association between a prompt and the generated concept. The results for KRS are presented in Tab. 3. **We aim for the Knowledge Recovery Score to serve as a practical benchmark that helps the community assess and improve unlearning effectiveness.**

---

> > > ### Author Response · Authors · 2025-11-27
> > >
> > > Thank you again for your comments. In the rebuttal, we addressed concerns about applications to modern architectures and expanded on the concepts of short- and long-term forgetting. Do you have any further remarks or concerns we could still address? We would be happy to improve the paper based on your feedback.

---

### Meta-Review · Area_Chair_7jL7 · 2026-01-07

**Summary:**

A primary concern is that the paper is largely diagnostic and empirical rather than methodological. While the Memory Self-Regeneration task usefully exposes weaknesses in existing unlearning methods, several reviewers noted that the proposed recovery strategy (MemoRa) is a straightforward combination of existing tools (DDIM inversion, interpolation, LoRA) without algorithmic novelty. The authors frame this simplicity as intentional, but for some reviewers this limits the paper's perceived contribution, especially for a venue expecting deeper technical advances.

Another major issue is the lack of strong theoretical or mechanistic explanation for why residual knowledge persists after unlearning. Although the rebuttal added representation-level analyses and metrics (FID, PCA visualizations, Knowledge Recovery Score), these remain empirical proxies rather than a principled account of where or how forgotten knowledge is stored. Reviewers felt that the short-term vs. long-term forgetting distinction, while appealing, still offers limited predictive or explanatory power.

Reviewers also questioned the paper's prescriptive value for unlearning research. The work convincingly shows that many unlearning methods are brittle, but provides only high-level guidance on how future methods should be designed to resist self-regeneration. As a result, the paper identifies a vulnerability more clearly than it advances concrete solutions.

Finally, although concerns about generalization beyond Stable Diffusion 1.4 were partly mitigated by added FLUX experiments, it remained cautious about how broadly the conclusions apply across architectures and settings.

Taken together, the paper is seen as a useful and well-executed empirical study that highlights an important gap in current unlearning evaluations, but one whose limited theoretical depth and incremental contribution make it hard to support an acceptance recommendation.

**Reviewer Concerns:**

Several reviewer concerns were substantially addressed in the rebuttal. Most notably, questions about the short-term vs. long-term forgetting distinction were reasonable and well-founded, and the authors responded with meaningful additions. The new representation-level analysis and the introduction of the Knowledge Recovery Score provide a clearer, more operational definition of these regimes. While still empirical, this goes beyond the original qualitative framing and strengthens the paper's central claim.

Other concerns were partially addressed but not fully resolved. Reviewers questioning the simplicity and lack of algorithmic novelty of MemoRa are correct that the method is a combination of existing components. The authors justify this as a deliberate design choice to expose vulnerabilities in unlearning rather than propose a new algorithm, but this does not fully resolve expectations of methodological innovation. Similarly, while ablations on reference images, inversion steps, and interpolation were added, the choice of specific components (e.g., LoRA vs. other recovery mechanisms) remains justified mainly on practicality rather than principle. Concerns about generalization beyond Stable Diffusion were partially addressed through added experiments, which seems not comprehensive enough in its current version.

Finally, several concerns remain largely outstanding. In particular, reviewers asking for deeper theoretical or mechanistic explanations of why unlearned diffusion models retain recoverable memory are only partially satisfied by empirical manifold analysis. The paper still lacks a formal account of where and how this knowledge is stored. In addition, questions about actionable guidance for unlearning method designers are only addressed at a high level.

**Reviewer Scores:**

Reviewer 7yCy raised many of the questions -- lack of mechanistic evidence for short- vs. long-term forgetting, no ablations, reliance on SD1.4, and limited guidance for unlearning developers. The rebuttal partially addressed several of these with new ablations, FLUX experiments, representation-level analysis, and the Knowledge Recovery Score. However, concerns about novelty and prescriptive guidance remain. This reviewer would likely keep their score 2.

Reviewer Axny was already positive and mainly asked about theoretical guarantees and continual unlearning. The authors' response acknowledged the lack of theory, clarified scope, and provided a reasonable future-work discussion on continual learning. Since the reviewer was already supportive and did not raise blocking concerns, their score would likely remain unchanged.

Reviewer 4P9M was skeptical about the short-/long-term framing, the role of interpolation, and whether FID meaningfully reflects recovery. The rebuttal addressed these points carefully: clarifying the definition of forgetting regimes, explaining FID's role, adding latent-space analysis, and providing justification and ablations for spherical interpolation. Importantly, this reviewer explicitly stated appreciation for the rebuttal and indicated they would update their score if possible. This suggests a clear upward revision, likely from a 2 to 4 or 6.

This reviewer was generally positive but questioned whether MemoRa reveals residual memory or just fast relearning, and asked for a quantitative metric. The rebuttal partially addressed these concerns by adding cross-subset recovery experiments and introducing the Knowledge Recovery Score. As a result, this reviewer would likely keep their score.

---

### Decision · Program_Chairs · 2026-01-26

Reject